# Revisiting the *Loa loa* microfilaremia thresholds above which serious adverse events may occur with ivermectin treatment

**Charlotte Boullé**[1,2]*, **Sébastien D. Pion**[1], **Jacques Gardon**[3], **Nathalie Gardon-Wendel**[4], **Joël Fokom Domgue**[5], **Joseph Kamgno**[6,7], **Cédric B. Chesnais**[1], **Michel Boussinesq**[1]

**1** UMI 233, Institut de Recherche pour le Développement, INSERM Unité, Université de Montpellier, Montpellier, France, **2** Department of Infectious and Tropical Diseases, Montpellier University Hospital, Montpellier, France, **3** Hydrosciences Montpellier, Université de Montpellier, Institut de Recherche pour le Développement, Centre National de la Recherche Scientifique, Montpellier, France, **4** Centre Pasteur, Yaoundé, Cameroon, **5** Department of Epidemiology, The University of Texas M.D. Anderson Cancer Center, Houston, Texas, United States of America, **6** Higher Institute for Scientific and Medical Research, Yaoundé, Cameroon, **7** Faculty of Medicine and Biomedical Sciences, University of Yaoundé I, Yaoundé, Cameroon

* c-boulle@chu-montpellier.fr (CB); michel.boussinesq@ird.fr (MB)

## Abstract

### Background

Loiasis was long deemed to be a benign condition, but individuals with high *Loa loa* microfilarial densities (MFD) are at risk of serious adverse events (SAEs) including encephalopathy following ivermectin (IVM) administration. The risk of marked AE or SAE is usually considered when MFD exceeds 8000 microfilariae (mf)/mL or 30,000 mf/mL, respectively. There are no international guidelines on the treatment of loiasis, resulting in a variety of practices worldwide for the treatment of infected individuals outside endemic areas. Our objective was to determine the probabilities of SAEs after IVM administration at the usual thresholds, and to refine those thresholds using individual characteristics such as age and sex.

### Methods

We used data from two clinical trials conducted in Cameroon where *L. loa* MFD were determined before IVM administration. The risk of SAE was modeled as a logistic function of age, sex, and MFD transformed as a first-order fractional polynomial.

### Principal Findings

SAEs probabilities were found to be <10[4] for MFD<2000 mf/mL, > 1‰ for MFD >8000 mf/mL, >1% for MFD>20,000 mf/mL, and >2.5% for MFD>30,000 mf/mL. We showed that specific categories may be at a higher risk of SAE than expected. Specifically, in order not to exceed 1% risk, the corresponding thresholds would be 18,000 mf/mL for females in the 31-40 age group; 16,000 mf/mL for males in the 21-30 age group; 12,000 mf/mL for males in the 31-40 age group; and 19,000 mf/mL for males in the 41-50 age group.

**Data availability statement:** Anonymized data will be hosted on the IRD dataverse server https://doi.org/10.23708/4GVQIJ and its terms of use will be those in force on the hosting site."

**Funding:** This work was supported by a grant from the Bettencourt-Schueller Foundation (CCU-AH grant to CB). The funders had no role in study design, data collection and analysis, decision to publish, or preparation of the manuscript.

## Conclusions

Our study suggests that IVM should be used with caution for males or individuals in specific age categories with a high *L. loa* MFD. For these high risk groups, lowering the thresholds to 8000 mf/mL should be considered. The increased risk in males requires further investigation to understand the pathophysiological phenomena involved that are crucial to prevent and manage SAEs.

## Author summary

Loiasis, a parasitic infection caused by *Loa loa*, used to be considered a mild condition. However, individuals with high numbers of larval stages (microfilariae) in their blood (microfilarial density or MFD) can experience serious adverse events (SAEs), such as severe neurological complications, after taking drugs that kill the microfilariae (ivermectin or diethylcarbamazine). There is currently no international guideline for the treatment of loiasis, leading to varied practices worldwide. The objective of this paper was to operationalize the results of previous modeling, to provide turnkey information to prescribers on the risks incurred for a given MFD, sex and age, and to provide additional elements for reflection on these risks. Using data from clinical trials conducted in Cameroon, we found that the probability of SAEs for a given MFD level varies between age and sex categories. We identified that men and individuals in specific age categories might be at a higher risk than previously thought. For example, males aged 31-40 might be at risk at lower MFD thresholds than currently used. Our findings suggest that treatment guidelines should be adjusted to consider these factors, and that more research is needed to understand why males seem to be at higher risk for *L. loa*-related post-ivermectin SAEs.

## Introduction

Loiasis is a parasitic disease caused by the filarial nematode *Loa loa* whose distribution is restricted to Central Africa. The disease is well known for two hallmark signs, namely the so-called "Calabar swellings" (transient angioedema usually affecting the upper limbs) and "eyeworm" (migration of an adult worm under the eye conjunctiva), but it can also induce pathogenic processes in various vital organs [1]. In endemic regions, loiasis hampers ivermectin (IVM) mass drug administration (MDA) organized to eliminate onchocerciasis because subjects harboring high *L. loa* microfilarial densities (MFD) in the blood can develop serious adverse events (SAEs), including potentially fatal encephalopathies appearing 3-5 days after IVM treatment [2].

There are no international nor national official guidelines for the individual treatment of loiasis. The reasons for this are probably that loiasis is often regarded as a benign condition, and that treatment of cases seen in travel medicine consultations is often guided by the availability of the effective drugs. However, loiasis can be associated with complications, including severe renal conditions [3,4], and two studies have shown that it may be associated with excess mortality [5,6], although causality has not yet been established. Consequently, this disease should be considered as a pathological condition *per se* and treatment modalities should be clarified.

Currently, three antiparasitic drugs are used for the treatment of loiasis. Definitive cure classically requires the use of diethylcarbamazine (DEC), with several courses often necessary

to ensure success [7,8], through its macrofilaricidal activity (i.e., killing adult worms). Adverse reactions to DEC are frequent, affecting almost half of the patients treated for loiasis [9]. DEC can induce SAEs including extra-membranous glomerulonephritis, pleural effusion [9], and potentially lethal encephalopathies when administered to patients with microfilaremia [10–12]. As mentioned above, practitioners have often to face the problem of availability of the drug, as some countries (including countries where loiasis is endemic) do not have access to DEC. This was recently pointed out by Gobbi *et al* [13], who analyzed data from travel medicine clinics belonging to the European Network for Tropical Medicine and Travel Health, and found that DEC was unavailable in 48% of them (33/69) and required several days to be available in 11 others. IVM has been suggested as an alternative treatment, rapidly (<7 days) improving symptoms even in amicrofilaremic individuals [14], although it is reputed to lack activity on macrofilariae. Indeed, IVM achieves a rapid and marked decrease in *L. loa* MFD [15–20] and this effect is the cause of the SAEs reported during IVM MDA in onchocerciasis endemic areas where loiasis is also present [2,21–25].

Albendazole (ALB) has little effect on *L. loa* MFD when given at single dose or short regimens [26,27] but treatment at 200 mg b.i.d. (twice a day) for 21 days leads to a progressive decrease in MFD by 60% in 6 months, suggesting an action on the adult worms' fecundity and/or survival [28]. In another study involving 16 subjects, a longer course of ALB at higher dose (400 mg b.i.d. for 28 days) followed by a single dose IVM led to high cure rate (94%) after 6 months – disappearance of clinical manifestations and of blood microfilariae (mf) [29]. ALB can be useful in case of relapse after several courses of DEC or in patients with high MFD prohibiting the use of DEC or IVM [30]. However, two cases of encephalopathy following ALB treatment (200 mg b.i.d.) were reported recently in patients with high *L. loa* MFD (35,000 mf/mL for the case with fatal outcome, 15,000 mf/mL for the other), suggesting that this drug may also induce SAEs [31,32]. The first case of post-ALB neurological SAE was reported in a patient with a low *L. loa* MFD (152 mf/mL) who developed a reversible encephalopathy after ALB treatment (200 mg b.i.d.) but in that case a differential diagnosis could not be ruled out with certainty [33]. A case of meningoencephalitis developing within one month after treatment with mebendazole – another member of the benzimidazole therapeutic class – has also been reported (the *L. loa* MFD measured 30 days after treatment exceeded 100,000 mf/mL) [34]. Besides treatments with anthelmintics, apheresis has been used as soon as 1969 to lower *L. loa* MFD in patients with high MFD levels, but this technique is rarely used [35].

Recent large case series reporting treatment assigned to patients with imported loiasis in France [36–38], Italy [39], England [40], and Spain [41] reflect the striking heterogeneity of drug use between countries [42] and even between specialized travel medicine clinics belonging to a same European Network [13].

The commonly applied strategy in France and other European countries is mainly based on a graded strategy based on the MFD expressed in mf/mL as proposed by Boussinesq [43,44]:

(i)  amicrofilaremic: DEC at progressively increasing dose in three doses per day (15mg q.d., double each day until reaching 8-10/mg/kg *quaque die* [q.d., each day], over 3-4 weeks)

(ii)  <2,000 mf/mL: DEC (starting 3-6mg q.d., double each day until reaching 400 mg q.d., over 3-4 weeks [45]);

(iii)  2,000-8,000 mf/mL: IVM 150 µg/kg every 1-3 months then DEC;

(iv)  8,000-30,000 mf/mL: IVM 150 µg/kg under hospital surveillance OR albendazole 200 mg b.i.d. during 21 days;

(v)  >30,000 mf/mL: ALB (although efficacy data on such high MFD is lacking) OR apheresis.

To our knowledge, in the USA, authors suggest treating with DEC if MFD is below 2,500 mf/mL or 8,000 mf/mL depending on sources, and otherwise advise that the treatment is conducted by experienced specialist, with ALB for instance. The CDC also recalls that therapeutic abstention may be considered, probably on the basis of the low pathogenicity attributed to loiasis [46,47].

This study focuses on IVM which is often used before, or preferred to DEC due to its availability – when DEC is frequently unavailable – and efficacy in reducing MFD. While previous analyses [48] modeled the relationship between *L. loa* MFD and SAEs in the context of MDA programs, the current study focuses on refining risk stratification to guide clinical management. This analysis systematically includes age category as a covariate to account for potential confounders and provides detailed SAE probability estimates stratified by MFD cutoffs, age, and sex. The objective of this paper is to operationalize the results of previous modeling, to provide turnkey information to prescribers on the risks incurred for a given MFD, sex and age, and to provide additional elements for reflection on these risks.

## Materials and methods

### Ethics statement

Studies were approved by the Cameroon National Ethics Committee for Research in Human Health (N°D31/ECF/MSP/SG/DSC/SDSSP/SEE on August 28th 1995 and FWA IRB00001954 on December 31st 2004). A written informed consent was signed by all voluntary participants in the 2005 study. For the Lekie study supported by the Special Programme for Research and Training in Tropical Diseases (WHO-TDR) in 1995-1996, the Ministry of Public Health of Cameroon considered that the study area was coendemic for onchocerciasis and thus needed to be treated with IVM, and that therefore no consent form had to be signed by the participants.

### Study population

We used data from two clinical trials in Cameroon where *L. loa* MFD were measured before IVM treatment [21,49]. The first was conducted in 1995–1996 in the Lekie division (in the Centre Region), to evaluate the incidence of post-IVM SAEs in a loiasis-endemic area, and the *L. loa* MFD threshold above which SAEs can occur. The trial enrolled 17,877 subjects in 106 villages, of whom 14,676 were IVM-naive. Among them, 6415 were aged ≥15 years. For technical reasons, 865 blood smears could not be examined leaving 5,550 adult patients with *L. loa* MFD measured. The second was conducted in 2005 in the Lom-et-Djerem and Haut-Nyong divisions (in the East Region) to evaluate whether *Plasmodium* sp. was a risk cofactor for *L. loa*-related post-IVM SAEs. It enrolled 4,956 IVM-naive subjects aged ≥13 years in 74 villages.

### Parasitological examination

In both trials calibrated (50 μL) thick blood smears (TBS) were prepared with finger prick blood collected between 10:00 AM and 4:00 PM and examined to measure the *L. loa* MFD. The slides were dehemoglobinized and stained with Giemsa within 48 h, and then stored and examined for mf after 1–2 weeks. The TBS were read by experienced microscopists and individual MFDs expressed in mf/mL of blood.

### Treatment and monitoring of serious adverse events

In both trials, IVM was given orally to eligible individuals at the standard dose of 150 μg/kg used as part of the Community-Directed Treatment with Ivermectin campaigns targeting

onchocerciasis. Active surveillance procedures were set up to detect the occurrence of AEs during the 7 days following treatment, and to manage them.

## Classification of adverse events

AEs were classified as: (i) *mild* if patients complained of reactions without functional impairment and *marked* in case of limited functional impairment requiring partial assistance for only several days; both types are referred to as "AEs" hereunder; (ii) *serious non-neurological* if patients had a functional impairment requiring full-time assistance for at least one week without neurological sign; and (iii) *serious neurological* if they developed disorders of consciousness and objective neurological signs and were hospitalized for appropriate management. Both neurological and non-neurological SAEs are referred to as "SAEs" hereunder. These definitions correspond to gradation of AEs linked to IVM, which has been adapted by Merck & Co. Inc. from the definitions of the International Council for Harmonisation of Technical Requirements for Pharmaceuticals for Human Use and agreed on by the Mectizan Expert Committee (MEC) – Mectizan is the trade name of IVM donated by Merck & Co., Inc. to combat onchocerciasis and lymphatic filariasis and the MEC is an independent body of experts established to provide technical oversight of the Mectizan Donation Program (MDP).

## Statistical analyses

To model the occurrence of an SAE, we constructed bivariate models using age, sex, and pre-treatment *L. loa* MFD as covariables. A first-order fractional polynomial (FP1) transformation of the *L. loa* MFD was used [48].

$$Loa^{FP} = \ln\frac{Loa + 2.22216796875}{100000} + 3.658185875 \tag{1}$$

Age was incorporated as a categorical variable (≤20, 21-30, 31-40, 41-50, >50). This transformation was guided by the shape of the second-order FP polynomial (FP2) relationship of age with the occurrence of SAEs [48].

$$Age^{FP}{}_2 = 10 \times \frac{\ln\left(\dfrac{Age}{10}\right)}{Age} - 0.357006446 \tag{2}$$

The bivariate model using this categorical categorization had minimal difference in Akaike Information Criterion (AIC) when compared to the FP2 transformation, and was chosen to avoid the use of an equation for age and to provide more easily readable estimates for end-users. Multivariable logistic models with SAEs as the dependent variable and *L. loa* MFD, age, and sex as independent variables were compared using the AIC. Margins post-estimation commands in Stata 17.0 were used to obtain predicted probabilities after estimating odds ratios in logistic regression. Risk probabilities of SAEs at the usual *L. loa* MFD cut-offs were retrieved using a logistic model including only the *L. loa* MFD. The risk probabilities of SAEs for MFD of 0, 2,000, 5,000, 8,000, 10,000, 20,000, 30,000, 50,000, and 100,000 mf/mL, for each of the age and sex categories (that were both significant in the multivariate model) were then estimated.

All analyses were performed using Stata 17.0 (StataCorp, College Station, Texas, TX, USA).

## Results

### Description of the study population

A total of 10,506 individuals were enrolled in the studies conducted in the Center and East regions in Cameroon. Thirty-eight SAEs occurred including 2 neurological SAEs. Those who presented SAEs, were more likely to be males (79.0%, p<0.001), and had higher *L. loa* MFD (range: 5,620-182,400 mf/mL) (Table 1).

### Predicted probabilities of post-IVM SAEs according to the *L. loa* MFD

SAE risk probabilities were calculated to be below 0.0001% for MFD below 2,000 mf/mL, of 0.1% for an MFD of 8,000 mf/mL, of 1% for an MFD of 20,000 mf/mL, and of 2.8% for an MFD of 30,000 mf/mL (Table 2).

The other way around, using rounded percentages, and calculating the exact corresponding MFD gave the following correspondences: for 0.005% it was 2,410 mf/mL, for 0.1% it was 7,935 mf/mL, for 1% it was 19,890 mf/mL, and for 2.5% it was 28,815 mf/mL.

### Predicted probabilities of post-IVM SAEs according to the *L. loa* MFD and individual characteristics

Probabilities of developing an SAE for given *L. loa* MFD values corresponding to different thresholds were computed using a multivariate model with *L. loa* MFD, age, and sex as covariates.

$$P_{SAE} = \frac{1}{1 + e^{-(2.563691 \times Loa_{FP1} + 1.049903 \times Sex + \beta_n \times Age_{cat} - 11.55906)}} \tag{3}$$

**Table 1. Description of the study population.**

|  | All individuals (N=10,506) | Individuals who did not experience SAEs (n=10,468) | Individuals who presented SAEs (n=38) | p-value |
|---|---|---|---|---|
| Age (median, (IQR)) | 31 (19-50) | 31 (19-50) | 36 (30-48) | 0.0240 |
| Age (categorical) |  |  |  | <0.001 |
| 13-20 | 3,147 (30.0%) | 3,145 (99.9%) | 2 (0.1%) |  |
| 21-30 | 1,984 (18.9%) | 1,975 (99.6%) | 9 (0.5%) |  |
| 31-40 | 1,523 (14.5%) | 1,510 (99.2%) | 13 (0.9%) |  |
| 41-50 | 1,428 (13.6%) | 1,422 (99.6%) | 6 (0.4%) |  |
| >50 | 2,424 (23.1%) | 2,416 (99.7%) | 8 (0.3%) |  |
| Sex |  |  |  | <0.001 |
| Female | 5,367 (51.1%) | 5,359 (99.9%) | 8 (0.2%) |  |
| Male | 5,139 (48.9%) | 5,109 (99.4%) | 30 (0.6%) |  |
| *L. loa* MFD |  |  |  | <0.001 |
| 0 | 7,534 (71.7%) | 7,534 (100.0%) | 0 (0%) |  |
| 1-2,000 | 1,425 (13.6%) | 1,425 (100.0%) | 0 (0%) |  |
| 2,001-8,000 | 704 (6.7%) | 703 (99.9%) | 1 (0.1%) |  |
| 8,001-30,000 | 599 (5.7%) | 597 (99.7%) | 2 (0.3%) |  |
| 30,001-50,000 | 133 (1.3%) | 126 (94.7%) | 7 (5.3%) |  |
| >50,000 | 111 (1.1%) | 83 (74.8%) | 28 (25.2%) |  |

In column 2, percentages are column percentages, and in columns 3 and 4, row percentages. SAE: serious adverse event; IQR: interquartile range; MFD: microfilarial density.

p-values were obtained using the Mann-Whitney (age), or chi-2 tests.

with:

$$\beta_n \begin{cases} \beta_0 = 0 \\ \beta_1 = 1.102808 \\ \beta_2 = 1.966523 \quad for \\ \beta_3 = 0.6945951 \\ \beta_4 = .4595063 \end{cases} \begin{cases} Age_{cat_0}\,[<20] \\ Age_{cat_1}\,[21-30] \\ Age_{cat_2}\,[31-40] \\ Age_{cat3}\,[41-50] \\ Age_{cat_4}\,[>50] \end{cases}$$

with $Age_{cat} = 0$ for reference category [<20] and $Age_{cat} = 1$ otherwise, $Sex = 0$ for females and $Sex = 1$ for males. $Loa_{FP1}$ is defined in equation (1) in the Material and methods section.

The regression table with parameters and their confidence intervals are given in **Supporting Information** S1 Table.

Results are presented in Table 3 and Fig 1.

For some specific sex and age categories, the rounded threshold to reach a predicted risk >2.5% (corresponding to the predicted overall risk when MFD exceeds 30,000 mf/mL) was lower than 30,000 mf/mL. Specifically, this threshold was 26,000 mf/mL (19,000-54,000) for females in the 31-40 age group; 24,000 mf/mL (18,000-41,000) for males in the 21-30 age group; 17,000 mf/mL (13,000-28,000)for males in the 31-40 age group; and 28,000 mf/mL (21,000-60,000) for males in the 41-50 age group. Considering the >1% predicted-risk (corresponding to the predicted overall risk when MFD exceeds 20,000 mf/mL) led to the same results, with threshold being of 18,000 mf/mL (13,000-42,000) for females in the 31-40 age group; 16,000 mf/mL (12,000-32,000) for males in the 21-30 age group; 12,000 mf/mL (9000-22,000) for males in the 31-40 age group; and 19,000 mf/mL (14,000-49,000) for males in the 41-50 age group.

## Discussion

Our risk estimates are overall higher than expected from the number of SAEs reported to MDP. From 1990 to 2017, more than 500 cases of characteristic post-IVM encephalopathy, including approximately 60 fatal cases, were reported to the MDP [50]. Detailed information has been published on the 65 "probable" or "possible *L. loa* encephalopathy temporally related to treatment with IVM" reported between 1989 and 2001 [51]. Vinkeles *et al.*

Table 2. **Predicted probabilities of post-IVM SAEs at various *L. loa* MFD cutoffs.**

| *L. loa* MFD (mf/mL) | Probability of post-IVM SAE | 95%CI |
|---|---|---|
| 0 | $<10^{-11}$% | $[0-<10^{-11}\%]$ |
| 2,000 | $<10^{-4}$% | $[0-<10^{-4}\%]$ |
| 5,000 | 0.03% | [0; 0.08%] |
| 8,000 | 0.10% | [0; 0.22%] |
| 10,000 | 0.18% | [0; 0.37%] |
| 20,000 | 1.01% | [0.32; 1.71%] |
| 30,000 | 2.76% | [1.39; 4.13 %] |
| 50,000 | 9.31% | [6.16; 12.45%] |
| 100,000 | 36.99% | [25.21; 48.76%] |

Green indicates probabilities of SAE below 0.005%, white indicates probabilities between 0.005% and 0.1%, yellow indicates probabilities between 0.1% and 1%, orange indicates probabilities between 1% and 2.5%, red indicates probabilities greater than 2.5%. MFD: microfilarial density; 95%CI: 95% confidence intervals

[52] estimated that in 1995, there were 122,300 individuals with MFD ≥20,000 mf/mL in MDA regions, corresponding to ~1,200 expected SAEs annually based on our estimates that SAE-risk above 20,000 mf/mL is close to 1%. Furthermore, repeated IVM treatments significantly reduce MFD, where only 1 out of 57 individuals remained above the 20,000 mf/mL threshold after one year [16], further diminishing the at-risk population for the next IVM round. In addition, repeated CDTI probably prevent the development of hyper-microfilaremia in individuals who still have MFDs below the risk threshold. While the discrepancy is acceptable from a programmatic perspective, the individual risk estimates remain plausible. The number of SAEs reported to the MDP is probably underestimated given the flaws in the AE surveillance system in some countries, particularly in remote areas where TBS are not always prepared to assess the post-treatment MFD required to characterize "probable" or "possible" cases, and before the implementation of community-directed treatment with IVM (CDTI), that allowed improved case identification and management over time.

The two patients who experienced neurological SAEs included in the study conducted in the Lekie Division in 1995-1996 harbored respectively 50,520 and 152,940 mf/mL and eventually recovered [21]. Six other cases of *Loa*-related post-IVM neurological SAE

**Table 3. Predicted probabilities of SAEs depending on sex, pre-treatment *L. loa* MFD and age.**

**a) Females**

| *L. loa* MFD (mf/mL) | Age category | | | | |
|---|---|---|---|---|---|
| | ≤20 | 21-30 | 31-40 | 41-50 | >50 |
| 0 | 0.00% | 0.00% | 0.00% | 0.00% | 0.00% |
| 2000 | 0.00% | 0.00% | 0.00% | 0.00% | 0.00% |
| 5000 | 0.01% | 0.02% | 0.04% | 0.01% | 0.01% |
| 8000 | 0.02% | 0.05% | 0.12% | 0.03% | 0.03% |
| 10,000 | 0.03% | 0.09% | 0.22% | 0.06% | 0.05% |
| 20,000 | 0.18% | 0.55% | 1.29% | 0.36% | 0.29% |
| 30,000 | 0.51% | 1.53% | 3.56%* | 1.02% | 0.81% |
| 50,000 | 1.88% | 5.44%* | 12.02%* | 3.69% | 2.94% |
| 100,000 | 10.15% | 25.39%* | 44.67%* | 18.45%* | 15.17%* |

**b) Males**

| *L. loa* MFD (mf/mL) | Age category | | | | |
|---|---|---|---|---|---|
| | ≤20 | 21-30 | 31-40 | 41-50 | >50 |
| 0 | 0.00% | 0.00% | 0.00% | 0.00% | 0.00% |
| 2000 | 0.00% | 0.00% | 0.01% | 0.00% | 0.00% |
| 5000 | 0.01% | 0.04% | 0.11% | 0.03% | 0.02% |
| 8000 | 0.05% | 0.15% | 0.35% | 0.10% | 0.08% |
| 10,000 | 0.09% | 0.27% | 0.63% | 0.18% | 0.14% |
| 20,000 | 0.52% | 1.55% | 3.59%* | 1.03% | 0.82% |
| 30,000 | 1.45% | 4.25%* | 9.53%* | 2.87% | 2.28% |
| 50,000 | 5.18% | 14.13%* | 28.07%* | 9.86%* | 7.96%* |
| 100,000 | 24.40% | 49.30%* | 69.76%* | 39.27%* | 33.82%* |

Green indicates probabilities of SAE below 0.005%, white indicates probabilities between 0.005% and 0.1%, yellow indicates probabilities greater than 0.1%, orange indicates probabilities greater than 1%, red indicates probabilities greater than 2.5%).

*indicates that the lower bound of the 95%CI for risk probability is >0 (p-value <0.05).

MFD: microfilarial density

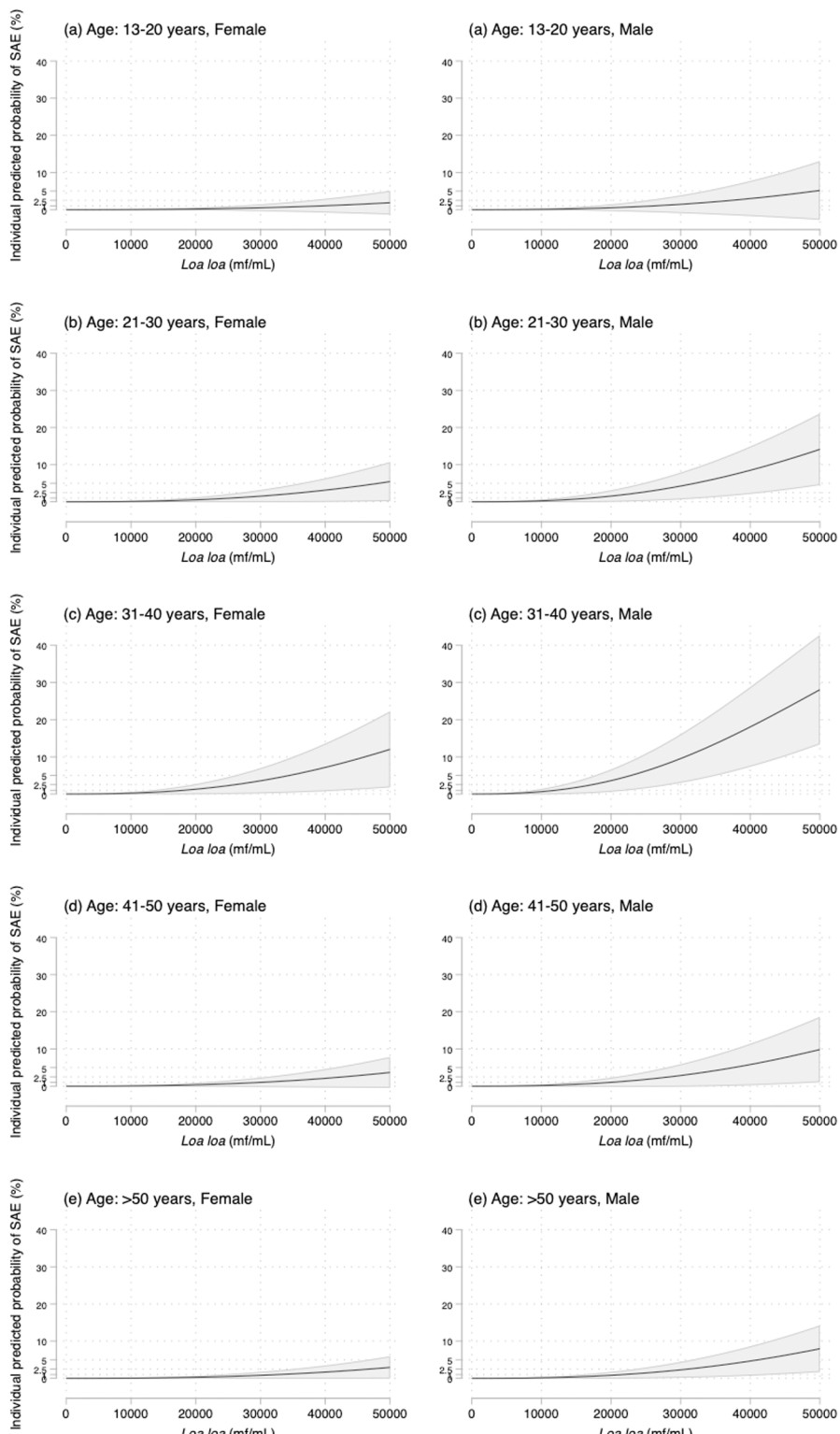

**Fig 1. Predicted probabilities (black line) with their 95% confidence interval (grey area).**

with known pre-treatment *L. loa* MFD have been reported, and all of them had an MFD exceeding 100,000 mf/mL: 162,920 mf/mL [23], 139,000, 109,000, 199,000 and 217,000 mf/mL [22]; and 120,000 mf/mL [53]. Six of the 8 neurological SAE cases were males, and 6/8 were aged between 20 and 40 years, the two others being 18 years old and 59 years old.

Besides the neurological SAEs, IVM can induce non-neurological SAEs and so-called "marked" AEs [21] which are associated with functional impairment for several days and also have a negative impact on the therapeutic coverages achieved during IVM MDA in loiasis endemic areas [54]. Actually, it is believed that there is a continuum in the severity between mild, marked, serious non-neurological and serious neurological adverse events [55], reflecting the importance of understanding the underlying mechanisms in order to develop prevention or alternative strategies.

The reasons why middle-aged males are more at risk to develop post-IVM SAEs for a given *L. loa* MFD are unclear. Pathophysiological mechanisms associated with these events are not fully understood. First, it has been debated as to whether or not they were solely attributable to infection with *L. loa* [56,57]. Pharmacovigilance studies highlighted that neurological SAEs may occur after IVM treatment for scabies, acarodermatitis, and strongyloidiasis, outside of *L. loa*-endemic areas [58,59] shedding light on various mechanisms that may be involved. In the absence of *L. loa* infection, SAEs can be due to IVM toxicity, through (i) accidental or intentional overdosing; (ii) concomitant use of drugs inhibiting CY3A4 (drug-drug interactions) or (iii) host susceptibility similar to that reported in several dog breeds such as collies or Australian shepherds. In these breeds, mutations of the *ABCB1/mdr-1* gene is frequent, causing a deficiency in the blood-brain barrier, penetration of IVM into the brain and binding of IVM to gamma-aminobutyric acid (GABA) receptors leading to a typical clinical picture, including lethargy, hypersalivation, ataxia, coma, mydriasis, clinical blindness, and seizures [55]. Recently, the case of a 13-year-old boy who developed IVM toxicity (with coma, ataxia, pyramidal signs, and binocular diplopia) 2.5 hours after having received a single standard dose for scabies was described. In that case, genotyping showed that he was a compound heterozygote for two nonsense mutations [60]. Nevertheless, most of the post-IVM SAE cases occur in *L. loa* endemic areas, and IVM toxicity accounts probably for very few of them in these settings. Besides this, a pilot study did not find loss-of-function mutations in *mdr-1* in four individuals who had experienced a SAE, but nonetheless found that two of them had mdr-1 polymorphisms [61].

Pathology examination was performed in a single case of post-IVM encephalopathy but as the subject died 54 days after IVM intake, the pathological findings are difficult to interpret [62]. To investigate the mechanisms associated with post-IVM encephalopathy, Wanji *et al* [63] conducted a study on 12 splenectomized baboons experimentally infected with *L. loa*. Once the *L. loa* MFD were deemed high enough, nine animals (with MFD ranging between 10,080 and 124,700 mf/mL) were treated with IVM. They developed various clinical manifestations: restlessness, fever, increase in respiratory and pulse rates, body rashes, gland pain, pinkish ears, swollen face, conjunctival hemorrhages, muscle aches, and loss of appetite – the animal with the highest MFD died 5 hours after treatment. Post-mortem examination revealed hemorrhages in various organs including the brain (similar to what can be seen in post-DEC encephalopathy), the lungs and the heart, with degenerating mf in small vessels associated with fibrin deposition and extravascular erythrocytes, and eosinophilic infiltration in tissues. Although the small sample size did not allow for multivariate analysis, there was an increase in the number of mf in the brain and kidneys as compared to those found in the three untreated controls, that was further increased in the two animals which had steroid co-administration but had higher initial MFD, and decreased in those three which had aspirin

co-administration. A study where a sample of 10 patients with MFD >30,000 mf/mL had lumbar puncture prior to IVM treatment showed that CSF was mf-free before treatment and that live mf were seen after treatment initiation, in CSF as well as in urine samples along with hematuria [23]. The frequent presence of sub-conjunctival hemorrhages associated with SAEs [64,65] and of retinal hemorrhages similar to those seen in cerebral malaria [64] are in favor of microvascular damages.

One of the questions that our study raises is the higher risk of SAEs in males, as compared with females in the same age and MFD categories. It may relate to unmeasured extrinsic cofactors (for instance alcohol intake, or tobacco smoking) or be linked to intrinsic differences and provide a clue to deciphering the mechanisms of toxicity. The influence of sex hormones and steroids hormone, which have in addition a circadian cycle deserves to be investigated. Other possibilities have been explored such as the influence of coinfections (*Mansonella perstans* [48], *Plasmodium* sp. [49]). Similar sex imbalance in other helminthic diseases with neurological manifestations have been evidence for instance for baylisascaridiasis [66], or gnathostomiasis [67], that also involve subcutaneous swellings and can causes fatal neurological disorders. The sex ratio for the latter has been suggested to be driven by different exposure between genders, although seroprevalence in Thailand can reach 63% [68] suggesting wide exposure.

Our study provides refined estimates of the MFD levels associated with the individual risk of SAE after IVM treatment. They might help guide individual treatment of microfilaremic individuals, both for imported loiasis or for the management of the disease in endemic areas where single dose treatment may be more feasible than bi-daily three-week long treatment with ALB. We did not use *M. perstans* MFD as a covariate because it was found to be only marginally significant in previous analyses [48] and our objective was to provide easily applicable data to guide treatment of loiasis outside of the endemic area.

The main limitations of our study are the lack of data on possible cofactors that could partially explain the gender-related excess risk, which is nevertheless consistent with previously published data. The overall strategy for prevention and management of post-IVM SAEs remains poorly studied, and is hampered by the lack of knowledge about the pathophysiological mechanisms. Although there is a risk of over-fitting due to the paucity of data, all available data was used in the present study in order to guide treatment administration. Indeed the data were collected exclusively in Cameroon, but we assume that the findings are likely generalizable to other endemic regions – or to patients outside endemic areas but who contracted the disease in Central Africa – but differences in clinical practices and on other comorbidities between countries may limit direct generalization.

Up to now, although used quite frequently, corticosteroid therapy is not recommended as it can precipitate decubitus complications depending on the local context which can themselves precipitate death in comatose patients. Its use in settings where nursing can be provided adequately needs definitively to be studied. Preliminary data on the baboon model has been obtained on the use of antithrombotic medication with aspirin, but this should be studied in depth before considering translation to human medicine in view of the benefit-risk balance giving the findings of tissue hemorrhages and vascular damages. Other adjuvant such as preventive anticoagulation should be studied. ALB was not the most widely used therapeutic in published series, and recent cases highlight that hypermicrofilaremic individuals receiving ALB should be carefully monitored, possibly in a hospital setting, until an antifilarial drug with ideal safety and efficacy profiles is available.

Other thresholds often used in the literature are 30,000 mf/mL (above which there is an increased risk of non-neurological SAEs) and 8,000 mf/mL (above which there is an increased risk of marked AEs).

Our results indicate that these thresholds for individual treatment guidance correspond to mean risks of SAEs around 0.1% for 8000 mf/mL, 1% for 20,000 mf/mL, and around 2.5% for 30,000 mf/mL. We also showed that specific categories may be at a much higher or sometimes lower risk of SAE than expected, when considering the individuals' age and sex. The fact that, for a given *L. loa* MFD, middle-aged males are at higher risk of SAE than other categories of the population had been reported previously [21,65]. The present study confirms this and leads us to propose that risk thresholds and individual treatment strategy be refined according to the age and sex of the patient.

Aligning with the threshold retained in the Test and Not Treat strategy relying on the LoaScope in endemic areas that allows exclusion of individuals with an estimated MFD≥20,000 mf/mL, corresponding to a global risk of 1% in our estimations, and to match with the categories that are routinely used, we would advise that the threshold for IVM use in *L. loa* microfilaremic patients is lowered down to 8000 mf/mL for males between 20 and 50 years, or females in the 30-40 age group. To simplify therapeutic recommendations, the threshold of 8000 mf/mL could thus be recommended for all individually treated patients (i.e., with TBS available). Further studies should investigate the correlation and variability between the LoaScope and the TBS to guide programmatic recommendations. Individuals harboring MFD above those thresholds should perhaps receive ALB or be considered for apheresis. New treatment options are currently being studied that could also constitute a valuable alternative, such as levamisole [69,70]. Given the lack of hindsight on safety of ALB at higher dosages in loiasis subjects (400 mg b.i.d. for 28 days, as used in some centers), and the recent concerns raised by two cases of SAEs in patients treated with ALB [31,32], it seems wise to keep with the protocol used by Klion *et al*, i.e., 200 mg b.i.d. during 21 days, and repeat if necessary [28]. Such a level of caution can be acceptable given the rarity of the disease outside endemic areas and the balance between benefits and risks.

## Conclusions

This study encourages to consider even more carefully the use of IVM especially in adult males, where it appears reasonable to lower the threshold until which it can be used to 8000 at most.

## Supporting information

**S1 Table. Multivariable logistic regression coefficients for probabilities of post-IVM SAEs according to the L. *loa* MFD and individual characteristics.**
(DOCX)

## Financial disclosure

This work was supported by a grant from the Bettencourt-Schueller Foundation (CCU-AH grant to CB). The funders had no role in study design, data collection and analysis, decision to publish, or preparation of the manuscript.

## Acknowledgments

We thank the participants of the clinical trials.

## Author contributions

**Conceptualization:** Cédric B Chesnais, Michel Boussinesq.

**Data curation:** Sébastien D Pion, Jacques Gardon, Nathalie Gardon-Wendel, Joël Fokom Domgue, Joseph Kamgno, Michel Boussinesq.

**Formal analysis:** Charlotte Boullé, Cédric B Chesnais.

**Investigation:** Jacques Gardon, Nathalie Gardon-Wendel, Joël Fokom Domgue, Joseph Kamgno, Michel Boussinesq.

**Methodology:** Charlotte Boullé, Sébastien D Pion, Cédric B Chesnais, Michel Boussinesq.

**Project administration:** Michel Boussinesq.

**Software:** Charlotte Boullé, Cédric B Chesnais.

**Supervision:** Michel Boussinesq.

**Validation:** Cédric B Chesnais, Michel Boussinesq.

**Visualization:** Charlotte Boullé.

**Writing – original draft:** Charlotte Boullé.

**Writing – review & editing:** Charlotte Boullé, Sébastien D Pion, Jacques Gardon, Nathalie Gardon-Wendel, Joël Fokom Domgue, Joseph Kamgno, Cédric B Chesnais, Michel Boussinesq.

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
