## [Decision Letter · Decision Letter 0]

22 Nov 2024

PNTD-D-24-01553Revisiting the Loa loa microfilaremia thresholds above which serious adverse events may occur with ivermectin treatmentPLOS Neglected Tropical DiseasesDear Dr. Boullé, Thank you for submitting your manuscript to PLOS Neglected Tropical Diseases. After careful consideration, we feel that it has merit but does not fully meet PLOS Neglected Tropical Diseases's publication criteria as it currently stands. Therefore, we invite you to submit a revised version of the manuscript that addresses the points raised during the review process. Please submit your revised manuscript within 60 days Jan 21 2025 11:59PM. If you will need more time than this to complete your revisions, please reply to this message or contact the journal office at plosntds@plos.org. Please include the following items when submitting your revised manuscript:

* A rebuttal letter that responds to each point raised by the editor and reviewer(s). You should upload this letter as a separate file labeled 'Response to Reviewers '. This file does not need to include responses to any formatting updates and technical items listed in the 'Journal Requirements' section below.

* A marked-up copy of your manuscript that highlights changes made to the original version. You should upload this as a separate file labeled 'Revised Manuscript with Track Changes '. * An unmarked version of your revised paper without tracked changes. You should upload this as a separate file labeled 'Manuscript '. If you would like to make changes to your financial disclosure, competing interests statement, or data availability statement, please make these updates within the submission form at the time of resubmission. Guidelines for resubmitting your figure files are available below the reviewer comments at the end of this letter. We look forward to receiving your revised manuscript.

Kind regards,

Luc E. Coffeng, MD PhD

Academic Editor

Eva ClarkSection EditorPLOS Neglected Tropical Diseases 

Shaden Kamhawi

co-Editor-in-Chief

Paul Brindley

co-Editor-in-Chief

**Journal Requirements:**

At this stage, the following Authors/Authors require contributions: Charlotte Boullé, Sébastien D Pion, Jacques Gardon, Nathalie Gardon-Wendel, Joel Fokom Domgue, Joseph Kamgno, Cédric B Chesnais, and Michel Boussinesq. Please ensure that the full contributions of each author are acknowledged in the "Add/Edit/Remove Authors" section of our submission form.

- © on Line: 209.

- ® on Line: 189.

3) Please upload all main figures as separate Figure files in .tif or .eps format with a numerical order. For more information about how to convert and format your figure files please see our guidelines:

4) In the online submission form, you indicated that Data availability: Data will be made available upon reasonable request to Dr Michel Boussinesq (michel.boussinesq@ird.fr).. All PLOS journals now require all data underlying the findings described in their manuscript to be freely available to other researchers, either

- In a public repository

- Within the manuscript itself

- Uploaded as supplementary information.

5) Please amend your detailed Financial Disclosure statement. This is published with the article. It must therefore be completed in full sentences and contain the exact wording you wish to be published. Please ensure that the funders and grant numbers match between the Financial Disclosure field and the Funding Information tab in your submission form. Note that the funders must be provided in the same order in both places as well.

**Reviewers' Comments:**

Reviewer's Responses to Questions

**Key Review Criteria Required for Acceptance?**

**Methods**

-Are the objectives of the study clearly articulated with a clear testable hypothesis stated?

-Is the study design appropriate to address the stated objectives?

-Is the population clearly described and appropriate for the hypothesis being tested?

-Is the sample size sufficient to ensure adequate power to address the hypothesis being tested?

-Were correct statistical analysis used to support conclusions?

-Are there concerns about ethical or regulatory requirements being met?

Reviewer #1: Objectives:

The study concerns a re-analysis of previously collected data. The analysis is very similar to the one published previously by the same group (Chesnais et al, ref 49 in the current paper). This earlier analysis is not cited in the introduction and the rationale for redoing the analysis with only marginally different methods is not explained. The main difference between the current and previous analysis relates to how the analysis is framed (related to the treatment of Loa loa infection or related to SAE prevented by not treating Loa loa cases in MDA programmes for onchocerciasis). The authors now choose for a logistical model that includes age, while previously they preferred a logistical model that did not include age, although models including age were also presented in ref 49! I think the authors should explain better in the introduction why their earlier analyses are not sufficient for the purpose described in the current paper (and hence why the current paper is worth publishing) and how they are improving on the earlier analysis.

Statistical methods: The authors state how loa MFD was included in their previous logistic models (line 195-196), but not how they do it in the current paper (e.g. line 197 and 204-206). I assume that they are still included is as a continuous variable after FP1 transformation. Did they use the exact same transformation as before (and which transformation was that?) or did they retest all candidate transformations again? This should be stated explicitly. Also, the authors should clarify how they derived the SAE probabilities presented in tables 2 and 3. Did they use the logistic equations to calculate the risk at the given cutoffs for table 2 and table 3? Similarly, how did they estimate the safety threshold presented in the last paragraph of the results section? Were they simply red from the graphs in the supplementary information?

Reviewer #2: The methods are clear and appropriate.

Reviewer #3: The objectives, design, sample size, statistical analysis are appropriate.

**Results**

-Does the analysis presented match the analysis plan?

-Are the results clearly and completely presented?

-Are the figures (Tables, Images) of sufficient quality for clarity?

Reviewer #1: The presentation of results is somewhat confusing, as table 2 seems to be based on their previously reported analysis (ref 49). Table 3 and figure 1 are new, but parameters of the logistic regression equations are not provided. Please provide the equations with information on confidence intervals around parameter estimates. Table 3 and figure 1 seem to present the same information, where predicted probabilities are extracted from the graph at given cut-off values for table 3 (not a bad idea, as the figure panels are too small for readers to extract such numbers themselves).

It is not clear why in table 3 the cells are color-coded, while the color coding is in a separate column in table 2.

Figure 1: explain the interpretation of the black line and grey area in the caption.

Reviewer #2: The results address the research question and are supported by the figures.

Reviewer #3: Results are adequately presented

**Conclusions**

-Are the conclusions supported by the data presented?

-Are the limitations of analysis clearly described?

-Do the authors discuss how these data can be helpful to advance our understanding of the topic under study?

-Is public health relevance addressed?

Reviewer #1: The discussion section is too long and presents a lot of information that is not directly linked to the analyses performed. I recommend to shorten it considerably, maybe even cut down the number of words by one-third or so. Some limitations of the analysis are described. For me, the main limitation is the lack of data in general: inferences are based on data from only 2 studies with 38 SAE occurrences. There may be a risk of overfitting of the data. Another aspect to address: to what extent are results generalizable to other countries? A statement about the added value of the current analysis over the previously presented one (in ref 49) may be useful.

Reviewer #2: The conclusions section needs substantial work. Right now it does not hold together well. It is trying to cover way too much in the way of case studies for not only IVM but DEC and ALB-associated SAEs. As a result it dilutes the overall message of this paper and the reader quickly becomes confused as to the purpose of the manuscript. The Discussion would be greatly improved if it focused on the impact of IVM on SAEs (with no more than a paragraph each on how the results might differ for DEC and ALB, based on experiences documented to date).

Reviewer #3: The authoers indicate that their results indicate that these thresholds for individual treatment guidance correspond

434 to mean risks of SAEs around 0.1% for 8000 mf/mL and around 2.5% for 30,000 mf/mL. However this doesnt correspond at all with the evidence we have from the Ivermectin donation program where 5 billion tablets have been donated and very little SAE or AE have occurred. This cannot be explained by underreporting. The prevalence of AE and SAE would be much higher if their conclusion was to be true. That should at least be considered in the discussion.

**Editorial and Data Presentation Modifications?**

Reviewer #1: (No Response)

Reviewer #2: This paper needs careful review by a native English speaker. There are quite a few minor grammatical errors and awkward phrasings. Some of which are shared below:

Line 42: Errant symbol after the “%”

Line 49: Replace “them” with something more specific like, “these high risk groups”

Line 88: Remove “being”

Line 104: Rather than “during”, I suggest “for” or “over the course of”; what does b.i.d. stand for?

Line 138: The use of “recalls” doesn’t make sense here, unless this is just a personal communication (and if so the individual should be attributed).

Line 257: “bond” should be “bound”

Line 280: instead of “have also” should be “also have”

Line 281-282: Do you mean a continuum related to MFD? If so, please state explicitly.

Line 289: Missing an “and” before strongyloides.

Line 290: consider “shedding light on” instead of “putting the light on”

Line 290: should be “In the absence”

Line 334: should we “were noted”

Line 388: Should be “imbalance”

Line 391: “In driven” should be “driven”

Reviewer #3: (No Response)

**Summary and General Comments**

Reviewer #1: MINOR COMMENTS:

1) Line 76: does it only hamper elimination programmes for onchocerciasis, or also those for lymphatic filariasis?

2) Line 84: a causal relation between loiasis mf density and excess mortality has not been proven. The cited studies merely demonstrate that there is an association, but other factors could explain the association (e.g. spending much time in the forest may not only be associated with loiasis, but may also entail other risks that could lead to mortality; socio-economic status; etc). Rephrase this sentence.

3) Line 142: the paper is not about AEs, but about SAEs. Rephrase the objective accordingly.

4) Line 195/196: not necessary to say what was done in a previous study. Just explain what you do in the current study.

5) Line 225/226: can this information be moved to the methods section?

6) Line 226/227: remove "Using the MFD thresholds used in the literature dealing with L. loa related SAEs".

7) Line 308: prognostic ==> prognosis

8) Line 403: please start a new paragraph for a discussion on the limitations.

9) Line 416/417: "that surveillance after treatment of hypermicrofilaremic individuals with this drug should be strictly monitored" ==> unclear, consider rephrasing

Reviewer #2: This manuscript has an important message to relay about the differences in SAE risk related to IVM treatment among Loa loa infected individuals, based on age and sex. The title states that it is revisiting the thresholds related to SAEs due to IVM treatment; however, the manuscript spends a lot of time describing SAEs due to Albendazole and DEC. At the moment the main message of the paper is easily lost with all the description of specific cases and specific treatment strategies. I strongly suggest tightening up the paper and focusing more on the IVM results and the implication in the discussion.

Some additional more minor revision requests to strengthen the paper:

Lines 249-258: There is not sufficient explanation of how the exact thresholds reported here (e.g. 2,410 mf/ml) were calculated. Are these the values that represent the exact MFD corresponding to the risk of SAEs (eg, 0.005%)? If so, why is it that they are said to approximate the a priori thresholds (e.g. 2,000 mf/ml threshold) and not the other way around. I find this paragraph confusing.

Lines 263-367: For these MFD thresholds by age and sex at the 2.5% risk threshold, can you provide a CI around each threshold to give a sense of how stable the value is? I know you can get a rough sense from Figure 1, but it would be helpful to report them in a table.

Please provide a little more background for why the >2.5% risk threshold of 30,000 is singled out in the text here and not 20,000. Since the LoaScope is calibrated to 20,000 and it corresponds to a >1% risk threshold, it feels a little odd not to also consider this threshold.

Line 306-307: I’m confused why this states that the risk is ~50,000 mf/ml when throughout the manuscript it refers to a risk of >1% occurring at 20,000 mf/ml. Please rephrase this sentence to clarify what you consider “risk” to be (e.g. is it >5% chance SAEs)?

I note that the conclusion of the manuscript is to lower the threshold to 20,000 mf/mL; however, there is no mention of the fact that this recommendation aligns with the current threshold used for Test and Not Treat, which is the primary strategy by which individuals will be treated with IVM in Africa.

Reviewer #3: (No Response)

PLOS authors have the option to publish the peer review history of their article (what does this mean? ). If published, this will include your full peer review and any attached files.

**Do you want your identity to be public for this peer review?** For information about this choice, including consent withdrawal, please see our Privacy Policy .

Reviewer #1: No

Reviewer #2: No

Reviewer #3: No

**Figure resubmission:** While revising your submission, please upload your figure files to the Preflight Analysis and Conversion Engine (PACE) digital diagnostic tool, https://pacev2.apexcovantage.com/. PACE helps ensure that figures meet PLOS requirements. To use PACE, you must first register as a user. Registration is free. Then, login and navigate to the UPLOAD tab, where you will find detailed instructions on how to use the tool. If you encounter any issues or have any questions when using PACE, please email PLOS at figures@plos.org. Please note that Supporting Information files do not need this step. If there are other versions of figure files still present in your submission file inventory at resubmission, please replace them with the PACE-processed versions.**Reproducibility:** To enhance the reproducibility of your results, we recommend that authors of applicable studies deposit laboratory protocols in protocols.io, where a protocol can be assigned its own identifier (DOI) such that it can be cited independently in the future. Additionally, PLOS ONE offers an option to publish peer-reviewed clinical study protocols. Read more information on sharing protocols at https://plos.org/protocols?utm_medium=editorial-email&utm_source=authorletters&utm_campaign=protocols

---

## [Decision Letter · Decision Letter 1]

27 Feb 2025

PNTD-D-24-01553R1Revisiting the Loa loa microfilaremia thresholds above which serious adverse events may occur with ivermectin treatmentPLOS Neglected Tropical Diseases Dear Dr. Boullé, Thank you for submitting your manuscript to PLOS Neglected Tropical Diseases. After careful consideration, we feel that it has merit but does not fully meet PLOS Neglected Tropical Diseases's publication criteria as it currently stands. Therefore, we invite you to submit a revised version of the manuscript that addresses the points raised during the review process.

Please submit your revised manuscript within 30 days Mar 29 2025 11:59PM. If you will need more time than this to complete your revisions, please reply to this message or contact the journal office at plosntds@plos.org. Please include the following items when submitting your revised manuscript:

* A rebuttal letter that responds to each point raised by the editor and reviewer(s). You should upload this letter as a separate file labeled 'Response to Reviewers '. This file does not need to include responses to any formatting updates and technical items listed in the 'Journal Requirements' section below.

* A marked-up copy of your manuscript that highlights changes made to the original version. You should upload this as a separate file labeled 'Revised Manuscript with Track Changes '.

* An unmarked version of your revised paper without tracked changes. You should upload this as a separate file labeled 'Manuscript '. If you would like to make changes to your financial disclosure, competing interests statement, or data availability statement, please make these updates within the submission form at the time of resubmission. Guidelines for resubmitting your figure files are available below the reviewer comments at the end of this letter.

We look forward to receiving your revised manuscript.

Kind regards, Luc E. Coffeng, MD PhDAcademic EditorPLOS Neglected Tropical Diseases Eva ClarkSection EditorPLOS Neglected Tropical Diseases

Shaden Kamhawi

co-Editor-in-Chief

Paul Brindley

co-Editor-in-Chief

**Additional Editor Comments :** Dear authors, we are ready to accept the manuscript for publication after you have addressed a last few comments from one of the reviewers. Hopefully, this will be relatively little effort, so I will keep my eye out for the revised version and push it forward once you are ready. **Journal Requirements:**

1)  Please add the legend for your Supporting Table file after the references list.

2) Please note that your Data Availability Statement is currently missing the DOI/accession number of each dataset OR a direct link to access each dataset. If your manuscript is accepted for publication, you will be asked to provide these details on a very short timeline. We therefore suggest that you provide this information now, though we will not hold up the peer review process if you are unable.

**Reviewers' comments:** Reviewer's Responses to Questions

**Summary and General Comments**

Reviewer #1: I’d like to thank the authors for the careful consideration of all the comments that were made the reviews. I feel that the manuscript has strongly improved. However some comments remain:

-Line 85/86: please do add a remark that the association does not necessarily imply causality

-Line 129: explain abbreviation q.d.

-Introduction, last paragraph: although various treatments for loiasis are being introduced in the introduction, the paper is only about ivermectin. This should be mentioned in the last paragraph of the introduction. Also, some rationale should be provided as to whý this choice is made.

-Line 201: incomplete sentence

-Line 215: add explanation on how the various multivariable models are compared. (Based on AIC? Or deviance?)

-Line 223: add the term “multivariable” to clarify that you switch to discussing the multivariable models here (before it was about bivariate models)

-S1 Table: clarify in the table legend that these coefficients are for the multivariable model.

-Lines 281-291: only the coefficients for males aged 31-40 are significantly different from 0 (see Table S1). However, the text also emphasizes differences for 21-30 year olds and for 41-50 year olds. I am not sure whether that is justifiable. This should be corrected or, at a minimum, discussed.

-Lines 295-311: Please clarify at the start of this paragraph what point you are addressing. I guess this relates to the comment from reviewer 3 that your risk estimates are way higher than expected from the number of SAEs reported to MDP? It is now very hard for the reader to understand why you are bringing all this information up.

-Line 397: replace “extrapolation” by “generalization”

-Lines 425-426: do we have any idea what proportion of (individually treated) patients have such high mf counts? Will this change have important implications for practice? Also, do you recommend that this lowered threshold for ivermectin treatment is also applied in test and not treat programmes for onchocerciasis? It may be worthwhile to clarify this also in the conclusion.

PLOS authors have the option to publish the peer review history of their article (what does this mean? ). If published, this will include your full peer review and any attached files.

**Do you want your identity to be public for this peer review?** For information about this choice, including consent withdrawal, please see our Privacy Policy .

Reviewer #1: No

**Figure resubmission:**

 **Reproducibility:** To enhance the reproducibility of your results, we recommend that authors of applicable studies deposit laboratory protocols in protocols.io, where a protocol can be assigned its own identifier (DOI) such that it can be cited independently in the future. Additionally, PLOS ONE offers an option to publish peer-reviewed clinical study protocols. Read more information on sharing protocols at https://plos.org/protocols?utm_medium=editorial-email&utm_source=authorletters&utm_campaign=protocols

---

## [Editor Report · Decision Letter 2]

3 Mar 2025

Dear Boullé,

We are pleased to inform you that your manuscript 'Revisiting the Loa loa microfilaremia thresholds above which serious adverse events may occur with ivermectin treatment' has been provisionally accepted for publication in PLOS Neglected Tropical Diseases.

Best regards,

Luc E. Coffeng, MD PhD

Academic Editor

Eva Clark

Section Editor

Shaden Kamhawi

co-Editor-in-Chief

Paul Brindley

co-Editor-in-Chief

---

## [Editor Report · Acceptance letter]

Dear Boullé,

We are delighted to inform you that your manuscript, "Revisiting the Loa loa microfilaremia thresholds above which serious adverse events may occur with ivermectin treatment," has been formally accepted for publication in PLOS Neglected Tropical Diseases.

Best regards,

Shaden Kamhawi

co-Editor-in-Chief

Paul Brindley

co-Editor-in-Chief
